# Comparison of Placental Three-Dimensional Power Doppler Vascular Indices and Placental Volume in Pregnancies with Small for Gestational Age Neonates

**DOI:** 10.3390/jcm8101651

**Published:** 2019-10-11

**Authors:** Sue-Jar Chen, Chie-Pein Chen, Fang-Ju Sun, Chen-Yu Chen

**Affiliations:** 1Department of Medicine, Mackay Medical College, New Taipei City 252, Taiwan; sjchen0619@hotmail.com (S.-J.C.); cpchen@mmh.org.tw (C.-P.C.); 2Department of Obstetrics and Gynecology, Mackay Memorial Hospital, Taipei 104, Taiwan; 3Department of Medical Research, Mackay Memorial Hospital, Taipei 104, Taiwan; fjsun.b612@mmh.org.tw

**Keywords:** three-dimensional power Doppler, placental vascular indices, placental volume, small for gestational age

## Abstract

This prospective observational study aimed to compare the changes in placental vascular indices and placental volume using three-dimensional power Doppler (3DPD) ultrasound in pregnancies with small for gestational age (SGA) neonates. We enrolled 396 women with singleton pregnancies from September 2013 to June 2016. Placental vascular indices, including the vascularization index (VI), flow index (FI), and vascularization flow index (VFI), and placental volume were obtained using 3DPD ultrasound in the first and second trimesters. Of the enrolled women, 21 delivered SGA neonates and 375 did not. In the first trimester, the SGA group had a significantly lower mean FI (25.10 ± 7.51 versus 33.10 ± 10.97, *p* < 0.001) and VFI (4.59 ± 1.95 versus 6.28 ± 2.35, *p* = 0.001) than the non-SGA group. However, there was no significant difference in the placental volume between the two groups during the first trimester. In the second trimester, the SGA group also had a significantly lower mean FI (27.08 ± 7.97 versus 31.54 ± 11.01, *p* = 0.022) and VFI (6.68 ± 1.71 versus 8.68 ± 3.09, *p* < 0.001) than the non-SGA group. In addition, a significantly smaller placental volume was noted in the SGA group (104.80 ± 24.23 cm^3^ versus 122.67 ± 26.35 cm^3^, *p* = 0.003) than in the non-SGA group during the second trimester. The results showed that a decreased placental VFI occurred earlier than a decreased placental volume in SGA pregnancies.

## 1. Introduction

Small for gestational age (SGA) and fetal growth restriction (FGR) both describe a neonatal birth weight or estimated fetal weight less than the 10th percentile for gestational age (GA) and are common complications associated with perinatal hypoxia, asphyxia, and even mortality [1]. Furthermore, SGA has been associated with both an increased risk of adverse perinatal outcomes and lifelong consequences [2,3].

The mechanisms of SGA/FGR are multifactorial and may be caused by fetal, maternal, or placental factors of which placental insufficiency is known to be a leading cause. Microscopically, the maldevelopment of villous trees, smaller intervillous spaces, and lower diffusive conductance have been reported in SGA/FGR placentas [4]. Furthermore, deficient remodeling of the spiral arteries supplying the placenta can occur from as early as the first trimester, and the severity of placental dysfunction has been associated with neonatal birthweight [5]. On the other hand, substantial remodeling of a fully hemochorial placenta occurs during the end of the first trimester and the start of the second trimester, and events during this time may influence the placental volume [5]. Therefore, the early detection of placental vascular alterations or changes in placental volume may be helpful to predict SGA/FGR pregnancies. The early identification of SGA/FGR pregnancies can improve clinical outcomes, since it can prompt appropriate antenatal surveillance of the fetus, management, or the initiation of treatment strategies to preserve placental function [6,7].

Currently, two-dimensional ultrasound biometry is the main method used to detect FGR during routine prenatal visits. However, first trimester measurements have been reported to have limited accuracy in identifying an inappropriately growing fetus [8]. As the process of growth restriction starts before a pathologically small fetal size can be detected by sonographic biometry, research is needed to elucidate an appropriate method to differentiate between normal and growth-restricted fetuses early in pregnancy.

Three-dimensional power Doppler (3DPD) ultrasound is used to noninvasively measure vascular indices. The quantitative Doppler vascular indices used to evaluate vascular changes include the vascularization index (VI), flow index (FI), and vascularization flow index (VFI). These vascular indices are measured according to the relative proportions of color voxels and signal intensity within a volume of interest. The VI represents the tissue vascular density and is calculated according to the number of color voxels in a volume. The FI represents the average blood flow intensity and is calculated as the average color value of all the color voxels. The VFI is calculated as VI × FI and represents both vascularization and blood flow [9].

The results of studies regarding changes in these placental vascular indices in the first trimester in SGA/FGR pregnancies have been inconsistent [10,11]. In addition, studies assessing placental volume in the first trimester of SGA/FGR pregnancies have also reported inconsistent results [12,13,14,15]. Furthermore, limited data have been reported regarding follow-up assessments of 3DPD placental volume and placental vascular indices in the different trimesters of SGA pregnancies. Hence, this study aimed to compare 3DPD placental volume and vascular indices in the first and second trimesters between pregnant women with and without SGA neonates. To the best of our knowledge, this is the first study to investigate placental volume and vascular indices in SGA pregnancies through the first and second trimesters.

## 2. Methods

### 2.1. Study Population

This prospective observational study of singleton pregnancies was conducted from September 2013 to June 2016 at Mackay Memorial Hospital, a tertiary referral medical center in Taiwan. The enrolled pregnant women were divided into SGA and non-SGA groups. SGA was diagnosed as birth weight <10th percentile, based on the nationwide singleton birthweight percentiles in Taiwan [16]. All of the women underwent sonographic examinations during aneuploidy screening in the first trimester (11 0/7–13 6/7 gestational weeks) and anatomy screening in the second trimester (21 0/7–23 6/7 gestational weeks). In the first trimester, GA was determined according to the last menstrual period and fetal crown-lump length (CRL), whereas biparietal diameter (BPD), abdominal circumference (AC), and femur length (FL) were used to estimate the size of the fetus in the second trimester. Miscarriage, periviable preterm birth, and pregnancies complicated with fetal chromosomal or structural anomalies detected by karyotyping or sonographic examinations were excluded from this study. The Institutional Review Board of Mackay Memorial Hospital approved this study, and all personal identifiers were anonymized prior to analysis.

Maternal characteristics, including maternal age, body mass index (BMI), medical history, gravidity, parity, delivery method, medical history, and pregnancy-associated complications were collected. Neonatal outcomes, including GA at birth, birth weight, and Apgar scores were obtained after birth. Chronic hypertension during pregnancy was defined as hypertension (systolic blood pressure ≥140 mmHg and/or diastolic blood pressure ≥90 mmHg) first detected <20 gestational weeks. Gestational hypertension was defined as new-onset hypertension ≥20 gestational weeks without the presence of proteinuria. Preeclampsia was defined according to the diagnostic criteria of the American College of Obstetricians and Gynecologists as gestational hypertension combined with proteinuria or new signs of end-organ dysfunction [17]. Gestational diabetes mellitus (GDM) was defined according to the National Diabetes Data Group criteria [18].

### 2.2. Ultrasound Examination

All of the ultrasound scans were performed using a Voluson E8 ultrasound machine (GE Medical Systems, Zipf, Austria) with a 2–8 MHz transabdominal probe. The placental vascular indices (VI, FI, and VFI) and placental volume were automatically calculated using Virtual Organ Computer-aided AnaLysis (VOCAL^TM^) imaging software (GE Medical Systems, Zipf, Austria), and expressed on a scale of 0–100 (Figure 1).

The probe was placed along the alignment of the placenta to obtain the full volume, and the placental margin was outlined to mark the maximum area. We repeated this procedure six times after rotating the probe 30° each time around the axis. The ultrasound instrument settings were kept the same for the first trimester (frequency: low; flow resolution: mild 2; balance: 165; smoothing: 4/5; ensemble: 13; line density: 6; power Doppler map: 5; artifact suppression: on; line filter: 2; quality: normal; wall motion filter: low 1; pulse repetition frequency: 0.9 kHz) and the second trimester (frequency: low; flow resolution: mild 1; balance: 150; smoothing: 4/5; ensemble: 16; line density: 6; power Doppler map: 5; artifact suppression: on; line filter: 2; quality: normal; wall motion filter: low 1; pulse repetition frequency: 0.9 kHz) examinations. The pulsatility index (PI) of the uterine artery was then measured using a transabdominal probe. The uterine artery was identified at the point where it crossed the external iliac artery, and three consecutive cycles were attained and measured. The PI value of the uterine artery was calculated as the mean of the right and left PI values of the uterine artery. The same experienced operator (Chen C.Y.) performed all the scans to avoid inter-observer variability. In addition, all the placental volumes and vascular indices were measured twice in the first 20 women to determine intrarater reliability.

### 2.3. Statistical Analysis

SPSS version 21.0 (IBM, Armonk, NY, USA) was used for all statistical analyses. The chi-square or Fisher’s exact test was used for categorical variables, and the Student’s *t*-test or Mann–Whitney *U* test was used for continuous variables. Intraclass correlation coefficients (ICCs) were used to assess the intrarater reliability of the measurements of the placental volume and the vascular indices. Univariate and multivariate logistic regression analyses were used to investigate the individual parameters related to SGA. The Wilcoxon signed-rank test was used to evaluate changes in the placental vascular indices and the placental volume in the first and second trimesters. Due to the possibility of within-subject dependency with repeated measurements, we used a generalized estimating equation (GEE) model to evaluate the time effect for the placental vascular indices and placental volume in the SGA group and non-SGA group. Receiver operating characteristic (ROC) curves and the Youden index were used to assess the optimal cut-off value of each parameter to predict SGA, and the areas under the curves (AUCs) were derived. The continuous variables are presented as the mean ± standard deviation or median (interquartile range), and the categorical variables are presented as the number (percentage). The results of the logistic regression analyses are presented as odds ratios (ORs) and 95% confidence intervals (CIs). A *p* value < 0.05 was considered to be statistically significant.

## 3. Results

After excluding the cases of lost to follow-up (*n* = 7), miscarriage (*n* = 2, at 13 and 16 gestational weeks), periviable preterm birth (*n* = 2, at 21 and 22 gestational weeks), and pregnancies complicated with fetal chromosomal anomaly (*n* = 2, trisomy 21 and trisomy 18) and severe structural anomaly (*n* = 1, gastroschisis), 396 pregnant women were enrolled in this study, of whom 21 (5.3%) delivered SGA neonates. The neonatal outcomes and maternal characteristics are shown in Table 1 The rate of cesarean delivery in the SGA group was higher than that in the non-SGA group (52.40% versus 24.80%, *p* = 0.005). The incidence rates of chronic hypertension and preeclampsia were higher in the SGA group than in the control group (19.05% versus 2.40%, *p* = 0.003, and 23.81% versus 0.53%, *p* < 0.001, respectively). The mean delivery age in the SGA group was earlier than that in the non-SGA group (36.67 ± 2.37 weeks versus 38.69 ± 1.56 weeks, *p* = 0.001), and the birthweights were 2175.00 ± 426.91 g and 3139.18 ± 406.73 g in the SGA and non-SGA groups, respectively (*p* < 0.001). Furthermore, the 1-minute Apgar score in the SGA group was lower than that in the non-SGA group (8.57 ± 1.33 versus 9.21 ± 0.97, *p* = 0.005); however, there was no significant difference in the 5-minute Apgar score.

Table 2 shows comparisons of maternal serum markers, fetal biometrics, and results of ultrasound examinations between the SGA and non-SGA groups. There were no significant differences in the maternal serum markers, including pregnancy-associated plasma protein-A (PAPP-A) and free beta-human chorionic gonadotrophin (β-hCG), between the two groups. No significant difference in CRL was noted between the two groups in the first trimester. One possibility might be because we determined the GA according to the CRL in early pregnancy. Nevertheless, significant decreases in BPD, AC, and FL were noted in the SGA group in the second trimester. In addition, the FI and VFI were significantly lower in the SGA group compared to the non-SGA group in the first trimester (FI: 25.10 ± 7.51 versus 33.10 ± 10.97, *p* < 0.001; VFI: 4.59 ± 1.95 versus 6.28 ± 2.35, *p* = 0.001). However, no significant differences were noted between the two groups in the VI and placental volume. Significant decreases in the FI and VFI were also noted in the SGA group compared to the non-SGA group in the second trimester (FI: 27.08 ± 7.97 versus 31.54 ± 11.01, *p* = 0.022; VFI: 6.68 ± 1.71 versus 8.68 ± 3.09, *p* < 0.001); however, the placental volume was significantly decreased in the SGA group compared to the non-SGA group (104.80 ± 24.23 versus 122.67 ± 26.35, *p* = 0.003). In addition, there was still no significant difference in the VI between the two groups in the second trimester. Moreover, no significant differences were noted in the uterine artery PI between the two groups in either the first or the second trimester. The intrarater reliability of the placental volume and vascular indices was excellent (ICC > 0.90) (Table 3).

As shown in Figure 2, ROC curve analyses revealed that the optimal cut-off values (sensitivity and specificity) of the first trimester FI and VFI to predict SGA were 27.88 (0.60 and 0.81) and 5.68 (0.56 and 0.81), respectively, and that the AUCs for the first trimester FI and VFI were 0.72 (95% CI, 0.62–0.82) and 0.70 (95% CI, 0.60–0.80), respectively. The optimal cut-off values (sensitivity and specificity) of the second trimester VFI and placental volume to predict SGA were 7.59 (0.62 and 0.76) and 109.43 cm^3^ (0.66 and 0.76), respectively, and the AUCs for the second trimester VFI and placental volume were 0.71 (95% CI, 0.62–0.80) and 0.71 (95% CI, 0.59–0.84), respectively.

Table 4 shows the results after adjustments in the multivariate logistic regression analysis. Significant differences were observed in the first trimester VFI (adjusted odds ratio (aOR): 0.76; 95% confidence interval (CI): 0.60–0.96; *p* = 0.024), second trimester VFI (aOR: 0.78; 95% CI: 0.63–0.96; *p* = 0.021), and second trimester placental volume (aOR: 0.97; 95% CI: 0.95–0.99; *p* = 0.007). We further analyzed the results of standardizing the measures by one standard deviation change, and significant differences were still observed in the first trimester VFI (aOR: 0.43; 95% CI: 0.25–0.73; *p* = 0.002), second trimester VFI (aOR: 0.39; 95% CI: 0.21–0.74; *p* = 0.004), and second trimester placental volume (aOR: 0.43; 95% CI: 0.25–0.77; *p* = 0.004).

Box-and-whisker plots revealed significant increases in the medians of the VI, VFI, and placental volume in the second trimester compared to the first trimester in both groups; however, the median of the FI was not significantly different between the two trimesters in the SGA group and significantly decreased in the non-SGA group (Figure 3). Furthermore, we performed a GEE analysis to investigate changes in the placental vascular indices and placental volume according to the GA (Figure 4). The GEEs for the VI, FI, VFI, and placental volume versus GA in the SGA group were: VI = 9.640 + 0.757 (GA) (*p* < 0.001), FI = 22.022 + 0.232 (GA) (*p* = 0.281), VFI = 1.717 + 0.223 (GA) (*p* < 0.001), and placental volume (cm^3^) = (−28.233) + 5.998 (GA) (*p* < 0.001). The GEEs for those in the non-SGA group were: VI = 9.144 + 0.923 (GA) (*p* < 0.001), FI = 35.120 − 0.158 (GA) (*p* = 0.012), VFI = 3.075 + 0.249 (GA) (*p* < 0.001), and placental volume (cm^3^) = (−48.182) + 7.615 (GA) (*p* < 0.001). The results showed that the VI, VFI, and placental volume were positively correlated with GA in both groups and that the FI was not. In addition, the VI, FI, VFI, and placental volume were lower in the SGA group than in the non-SGA group during the first and second trimesters according to the GEEs.

## 4. Discussion

The results of this study demonstrated significantly lower placental FI and VFI values in the SGA group during the first and second trimesters. Data regarding changes in placental vascular indices in the first trimester of SGA/FGR pregnancies have been inconsistent [10,11]. Farina et al. performed a systematic review to study the relationship between placental vascular indices and SGA/FGR pregnancies and concluded that placental vascular indices were feasible only during the third trimester [10]. However, González-González et al. recently reported significantly lower values of all vascular indices in FGR pregnancies during the first trimester [11], which is consistent with our results except for the VI. Although not significant, there was a trend of a lower VI in the SGA group in our study, which may have been due to the limited number of SGA cases. On the other hand, previous studies have demonstrated that the FI, which represents blood flow intensity, is the most reliable vascular index due to lower intraplacental variability and higher intra- and inter-observer correlations [19,20]. We also found no significant difference in the FI between the first and second trimesters in the SGA group, implying that the FI is a reliable and stable index in SGA placentas.

Abnormal trophoblastic invasion and impaired remodeling process of the spiral arteries have been reported to lead to inadequate flow in uteroplacental circulation, resulting in SGA/FGR [5]. Collins et al. performed a sonographic study to examine the spiral arteries and found that the pulsatility and resistance indices of spiral artery jets were different in SGA pregnancies [21]. Lu et al. performed a morphometric analysis of stem villus arteries and found a higher vessel wall thickness/lumen ratio in SGA/FGR placentas [22]. These findings imply that placental blood flow intensity may be lower in SGA pregnancies, supporting our results of a significantly lower placental FI and VFI in the SGA group. In addition, deficient spiral artery remodeling of SGA/FGR placentas has been reported to occur from early pregnancy [5], and we also found that placental vascular changes could occur as early as the first trimester in the SGA group. Most previous studies have shown a decrease in placental vascular indices in SGA/FGR pregnancies after the second trimester, which is consistent with our results [19,23,24].

In this study, we revealed that a significant increase of placental volume from the first to the second trimester of pregnancy in both groups. De Paula et al. established nomograms of placental volume from 12 to 40 gestational weeks by three-dimensional ultrasonography and found that placental volume increases according to GA [25]. However, the results of previous studies on changes in placental volume during the first trimester in SGA/FGR pregnancies have been inconsistent [12,13,14,15]. In the current study, we found that the placental volume did not significantly change in the SGA pregnancies during the first trimester. One pathophysiology of SGA placentas is the underdevelopment of uteroplacental circulation followed by the recruitment of spiral arteries, which is influenced by the density of villous vascular networks. Subsequently, dysfunctional placental remodeling is affected from the beginning of the second trimester [5]. These mechanisms are consistent with our results of an earlier change in the placental vascular indices than the placental volume in the SGA pregnancies. Most previous studies have shown a decrease in placental volume in SGA/FGR pregnancies after the second trimester, which is consistent with our results [23,26,27].

Some studies have investigated the role of PAPP-A and free β-hCG during the development of the placenta in the first trimester and have reported that they are associated with SGA pregnancies [28]. However, a recent meta-analysis concluded that the predictive values of PAPP-A and free β-hCG are poor [29]. Some studies have reported that a higher uterine artery PI value may indicate impaired placentation and that this can be used to predict adverse pregnancy outcomes [30]; although a recent review concluded that the use of uterine artery PI as a single predictive test for FGR pregnancies had poor accuracy [31]. We did not find significant differences in PAPP-A, free β-hCG, or uterine artery PI between the SGA and non-SGA groups; however, we only enrolled a small number of SGA pregnancies.

We also found positive correlations between GA and the VI, VFI, and placental volume in this study but not FI. However, Hata et al. reported clear differences in the reference, range, and changes in the VI, FI, and VFI values during pregnancy [12]. The differences in the results may be due to differences in the volume measurements and settings of the ultrasound system. Whole placental volume and placental vascular sonobiopsy can be used to measure placental vascularization, and whole placental volume has been reported to be a more valuable technique compared to placental vascular sonobiopsy [32]. We measured the whole placental volume in the present study, and all of the parameters were measured using the same instrument settings by the same examiner, with excellent intraobserver agreement.

There are several limitations to this study. First, although the intrarater reliability of the 3DPD ultrasound examinations was excellent, we did not investigate interrater reproducibility. Second, we measured the whole placental volume, which is not easily measured in the third trimester, and thus we could not investigate the relationship between the 3DPD placental vascular indices and the placental volume in the latter stages of the SGA pregnancies. Third, because we did not investigate placental vascular indices and placental volume during the third trimester, we were unable to compare early and late FGR using 3DPD ultrasound. Finally, we conducted the sonographic examinations from the late first trimester, and the possibility that these observations might develop earlier could not be confirmed in this study.

## 5. Conclusions

Placental 3DPD vascular indices can provide an insight into placental vascularization in pregnancies with SGA neonates from early pregnancy. The placental VFI, rather than placental volume, can be used as a feasible sonographic marker in the first trimester to predict SGA. In addition, a decreased placental VFI occurred earlier than a decreased placental volume in the SGA pregnancies in this study. Earlier identification of SGA pregnancies may prompt appropriate antenatal surveillance of the fetus, management, or the initiation of treatment strategies to improve neonatal outcomes. Further studies with larger numbers of SGA neonates are warranted to confirm our findings and evaluate the possibility of using placental vascular indices for clinical applications with other parameters.

## Figures and Tables

**Figure 1 jcm-08-01651-f001:**
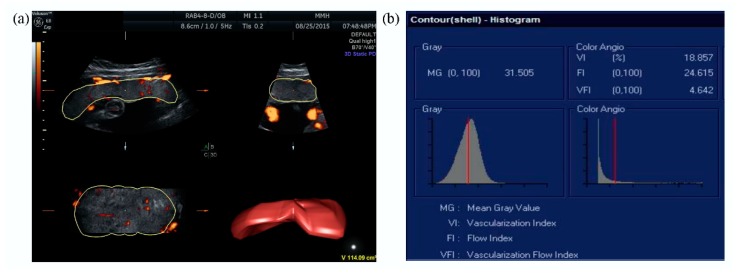
Placental vascular indices were assessed after evaluation of the whole placenta and calculated using Virtual Organ Computer-aided AnaLysis (VOCAL^TM^) imaging software: (**a**) placental volume measurement and (**b**) determination of placental vascular indices using three-dimensional power Doppler.

**Figure 2 jcm-08-01651-f002:**
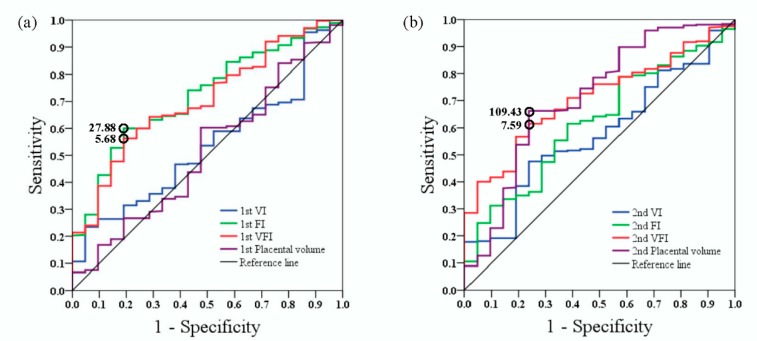
Receiver operating characteristic (ROC) curves of different parameters (VI, FI, VFI, and placental volume) in the (**a**) first and (**b**) second trimesters to predict SGA pregnancies. VI: vascularization index; FI: flow index; VFI: vascularization flow index. The optimum cut-off point (open circle) was defined as the closest point on the ROC curve to the point (*x*, *y*) = (0, 1), where *x* = 1-specificity and *y* = sensitivity.

**Figure 3 jcm-08-01651-f003:**
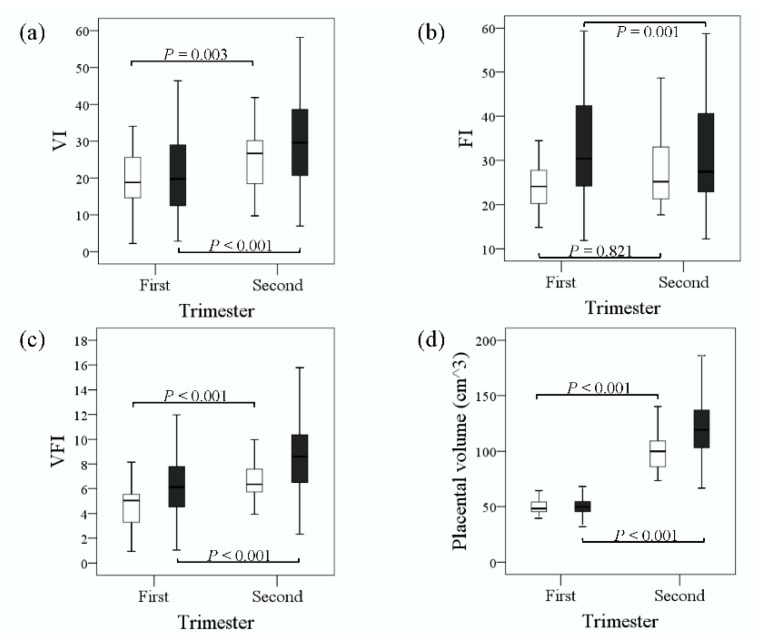
Box-and-whisker plots of (**a**) VI, (**b**) FI, (**c**) VFI, and (**d**) placental volume during the first and second trimesters in the SGA (□) and control (■) groups. VI: vascularization index; FI: flow index; VFI: vascularization flow index. Boxes show median and interquartile range, and whiskers represent 5th and 95th centiles.

**Figure 4 jcm-08-01651-f004:**
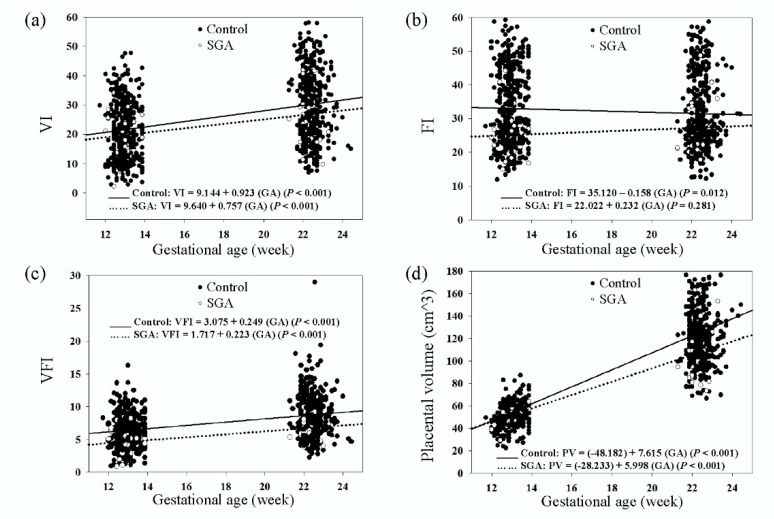
Generalized estimating equation analyses of (**a**) VI, (**b**) FI, (**c**) VFI, and (**d**) placental volume during the first and second trimesters in the SGA (∙∙∙) and control (—) groups. VI: vascularization index; FI: flow index; VFI: vascularization flow index; GA: gestational age; PV: placental volume.

**Table 1 jcm-08-01651-t001:** Maternal characteristics and neonatal outcomes.

	SGA (*n* = 21)	Non-SGA (*n* = 375)	*p*
Mother			
Age (years)	31.67 ± 3.75	31.96 ± 3.66	0.724
BMI (kg/m^2^)	21.65 ± 3.72	21.80 ± 3.62	0.849
Gravida	2.19 ± 1.33	2.02 ± 1.09	0.502
Para	0.62 ± 0.59	0.57 ± 0.65	0.740
Cesarean delivery	11 (52.40)	93 (24.80)	0.005 *
Chronic hypertension	4 (19.05)	9 (2.40)	0.003 *
Gestational hypertension	0	3 (0.80)	>0.999
Preeclampsia	5 (23.81)	2 (0.53)	<0.001 *
Type I DM	0	2 (0.53)	>0.999
Type II DM	0	5 (1.33)	>0.999
GDM	1 (4.76)	30 (8.00)	>0.999
Neonate			
Delivery age (weeks)	36.67 ± 2.37	38.69 ± 1.56	0.001 *
Birth weight (g)	2175.00 ± 426.91	3139.18 ± 406.73	<0.001 *
Apgar score			
1 minute	8.57 ± 1.33	9.21 ± 0.97	0.005 *
5 minutes	9.48 ± 0.87	9.69 ± 0.63	0.277

Continuous variables are presented as mean ± standard deviation and categorical variables as number (percentage). SGA: small for gestational age; BMI: body mass index; DM: diabetes mellitus; GDM: gestational diabetes mellitus. * *p* < 0.05.

**Table 2 jcm-08-01651-t002:** Maternal serum markers, fetal biometrics, and results of ultrasound examination between the SGA and non-SGA groups.

	SGA (*n* = 21)	Non-SGA (*n* = 375)	*p*
**First trimester**			
GA at examination (weeks)	12.90 ± 0.51	12.95 ± 0.44	0.600
PAPP-A (IU/L)	3.63 (5.14)	5.39 (4.73)	0.226
Free β-hCG (IU/L)	43.36 (28.37)	45.50 (38.90)	0.260
CRL (mm)	66.39 ± 6.88	67.66 ± 5.99	0.348
VI	19.32 ± 8.25	20.99 ± 9.63	0.436
FI	25.10 ± 7.51	33.10 ± 10.97	<0.001 *
VFI	4.59 ± 1.95	6.28 ± 2.35	0.001 *
Placental volume (cm^3^)	49.46 ± 8.22	50.34 ± 9.30	0.672
Uterine artery PI	1.69 ± 0.57	1.68 ± 0.44	0.939
**Second trimester**			
GA at examination (weeks)	22.24 ± 0.45	22.42 ± 0.49	0.089
BPD (cm)	5.31 ± 0.31	5.49 ± 0.23	0.001 *
AC (cm)	17.04 ± 0.94	17.68 ± 0.97	0.004 *
FL (cm)	3.70 ± 0.20	3.81 ± 0.19	0.007 *
VI	26.57 ± 9.54	29.94 ± 11.68	0.195
FI	27.08 ± 7.97	31.54 ± 11.01	0.022 *
VFI	6.68 ± 1.71	8.68 ± 3.09	<0.001 *
Placental volume (cm^3^)	104.80 ± 24.23	122.67 ± 26.35	0.003 *
Uterine artery PI	1.02 ± 0.42	0.93 ± 0.22	0.330

Variables are presented as mean ± standard deviation or median (interquartile range). SGA: small for gestational age; GA: gestational age; PAPP-A: pregnancy-associated plasma protein-A; β-hCG: beta-human chorionic gonadotrophin; CRL: crown-lump length; VI: vascularization index; FI: flow index; VFI: vascularization flow index; PI: pulsatility index; BPD: biparietal distance; AC: abdominal circumference; FL: femur length. * *p* < 0.05.

**Table 3 jcm-08-01651-t003:** Intraclass correlation coefficients of placental vascular indices and placental volume.

	ICC	95% CI	*p*
VI	0.983	0.958–0.993	<0.001 *
FI	0.929	0.821–0.972	<0.001 *
VFI	0.954	0.885–0.982	<0.001 *
Placental volume	0.961	0.903–0.985	<0.001 *

VI: vascularization index; FI: flow index; VFI: vascularization flow index; ICC: intraclass correlation coefficient; CI: confidence interval. * *p* < 0.05.

**Table 4 jcm-08-01651-t004:** Univariate and multivariate logistic regression analyses of the VFI and placental volume.

Variables	Univariate Logistic Regression			Multivariate Logistic Regression		
	Unadjusted OR	95% CI	*p*	Adjusted OR	95% CI	*p*
**First trimester**						
VFI	0.70	0.56–0.88	0.002 *	0.76	0.60–0.96	0.024 *
Placental volume	0.99	0.94–1.04	0.671	0.99	0.94–1.05	0.872
**Second trimester**						
VFI	0.74	0.61–0.90	0.003 *	0.78	0.63–0.96	0.021 *
Placental volume	0.97	0.94–0.99	0.003 *	0.97	0.95–0.99	0.007 *
**First trimester (per 1 SD)**						
VFI	0.43	0.25–0.73	0.002 *	0.43	0.25–0.73	0.002 *
Placental volume	0.91	0.59–1.41	0.671	0.99	0.65–1.52	0.969
**Second trimester (per 1 SD)**						
VFI	0.41	0.23–0.74	0.003 *	0.39	0.21–0.74	0.004 *
Placental volume	0.39	0.22–0.73	0.003 *	0.43	0.25–0.77	0.004 *

VFI: vascularization flow index; OR: odds ratio; CI: confidence interval; SD: standard deviation. * *p* < 0.05.

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
