# Peer review of "Comparison of Placental Three-Dimensional Power Doppler Vascular Indices and Placental Volume in Pregnancies with Small for Gestational Age Neonates"

_jcm, 2019, doi:10.3390/jcm8101651_

Round 1
Reviewer 1 Report
The scientific work is well structured, the argument is extremely interesting. The number of patients with SGA is low so I hope that in the future you could present an higher statistics. The English language needs minor revision by a native English speaker
Reviewer 2 Report
Table 4 and Figure 2 are the same. I suggest to present only Fig 2, which is more impressive.
The screening periode in first and second trimester was not only 5 days. In the Figure 4 is inaccurate.
Figure 3/d represents a phisiological state. The authors have to present box-plot analyses between SGA and non-SGA groups.
The article is very interesting, they allpy a new method (VOCAL) for research the SGA. The study contains data in the first and second trimester, it is unique in placental ultrasound investigations. well written, text clear and easy to read
Reviewer 3 Report
Overall, this is an important study that uses a prospective panel design to consider first trimester changes in placental blood flow and perfusion using 3-D ultrasound to examine early changes related to poor fetal growth. This work adds to a growing body of evidence that the placental changes ultimately limiting fetal growth occur early in pregnancy even if they are not detectable as changes in size until later. However, there are some issues that should be addressed and/or could strengthen the reported work:
The authors develop prediction criteria for early identification of SGA. In my view this is premature as this work adds to a growing body of evidence that early placental changes are associated with fetal growth, but to develop screening criteria seems to me to complicate their otherwise elegant analysis. A key concern that should be addressed is the determination of gestational age via 1st trimester ultrasound. Although it is not explicitly stated, I presume that this estimation was done at the same visit during which the placental measures were taken. This can create reverse causality, and may provide an explanation for the lack of observed differences in CRL in the first trimester. This possibility should be considered in the discussion. The units for the observed odds ratios were not provided. One presumes that these were per 1 unit change in the various placental vascular indexes. However, this is not explicit and given the observed differences in the scale of the various measures, they make it difficult to contextualize these changes. One could rectify this by either standardizing the measures and presenting the ORs for a 1 standard deviation change or by creating OR for an IQR change (i.e., the odds of SGA at the 75th percentile of the measure versus the odds of SGA at the 25th percentile. Either approach would improve the interpretability of these findings. Limited data were also presented about the specific methods of participant recruitment and the number of participants who were lost and or excluded. Finally, a minor comment is that the 1st trimester ultrasound was conducted late in the first trimester. The possibility that these observations might develop earlier could be considered in the discussion.
